# The SmartHabits: An Intelligent Privacy-Aware Home Care Assistance System

**DOI:** 10.3390/s19040907

**Published:** 2019-02-21

**Authors:** Andrej Grgurić, Miran Mošmondor, Darko Huljenić

**Affiliations:** Ericsson Nikola Tesla d.d., Krapinska 45, 10002 Zagreb, Croatia; miran.mosmondor@ericsson.com (M.M.); darko.huljenic@ericsson.com (D.H.)

**Keywords:** ambient intelligence, Internet of Things, ambient-assisted living, pattern recognition, anomaly detection

## Abstract

Many researchers and product developers are striving toward achieving ICT-enabled independence of older adults by setting up Enhanced Living Environments (ELEs). Technological solutions, which are often based on the Internet of Things (IoT), show great potential in providing support for Active Aging. To enhance the quality of life for older adults and overcome challenges in enabling individuals to achieve their full potential in terms of physical, social, and mental well-being, numerous proof-of-concept systems have been built. These systems, often labeled as Ambient Assisted Living (AAL), vary greatly in targeting different user needs. This paper presents our contribution using SmartHabits, which is an intelligent privacy-aware home care assistance system. The novel system comprising smart home-based and cloud-based parts uses machine-learning technology to provide peace of mind to informal caregivers caring for persons living alone. It does so by learning the user’s typical daily activity patterns and automatically issuing warnings if an unusual situation is detected. The system was designed and implemented from scratch, building upon existing practices from IoT reference architecture and microservices. The system was deployed in several homes of real users for six months, and we will be sharing our findings in this paper.

## 1. Introduction

Apart from the trend of population aging, which became one of the most significant social transformations of the 21st century, the number of single-person households is also on the rise. Current demographic trends will continue to put tremendous pressure on national healthcare, insurance, and economic sectors. Thus, it is of pivotal importance to provide new and innovative solutions. Provision of better care at the same or lower cost and delivering user-centric solutions with the help of the innovative ICT is one of the efforts aimed at solving this complex problem. Over the last several decades, care for older adults has vastly improved thanks to various technological advancements, but a lot still needs to be done. Technologies that help improve the quality of life of older adults can be categorized into three areas: safety, health and wellness, and social connectedness.

The ICT technology cannot, by itself, solve all the problems, but it has proven its usefulness in many cases. Much effort still has to be invested to gain the trust of the users and achieve wide-acceptance. The fear of technology, combined with the lack of experience and privacy issues, slows down the acceptance rate but the progress is noticeable. To enhance the quality of life for older adults and overcome challenges in enabling individuals to achieve their full potential in terms of physical, social and mental well-being, numerous Proof-of-Concept (PoC) systems have been built. However, one of the first findings following many PoCs (in different application areas) is that potential revenue streams are often too small with the business case being unclear. If the business value of the system does not exceed the costs of the whole system, including more than the technical components of the whole value chain needed for a service offering, new approaches have to be found and applied.

Systems that are non-invasive, self-adaptable (to user behavior), zero-touch, and easy to manage are still mostly missing. While older persons, especially ones living alone, still feel the technological frustration, their family members continue to worry and search for solutions that could help.

In this paper, we present the SmartHabits system, which was designed and implemented from scratch, building upon existing practices from IoT reference architecture and micro-services. The system was built in the company incubator following lean start-up methodology and iterative development to deliver a trial-ready prototype for testing by the real users. The main idea was to develop the system capable of answering the question “Is my loved one OK?” and to validate the concept in the real environment. The system is composed of three main parts: the Home-Sensing Platform (residing in the home of the older person living alone), the SmartHabits Expert System (residing in the Cloud, detecting anomalous situations in the home, and delivering notifications), and the underlying communication part. We also discuss the pilot setup and feedback gained from the successful pilot trials as well as some open challenges in the field.

This paper is organized as follows. In Section 2, a definition of the field and related work surveys are provided. Section 3 describes the SmartHabits system and gives details on system requirements and system architecture. Section 4 describes the pilot case together with targeted end-user groups, legal considerations, and the hardware used. Section 5 gives an extensive analysis of the results of the pilot study. In Section 6, the challenges and obtained results are discussed. Lastly, the conclusion of this work is given in Section 7.

## 2. Background

### 2.1. Definition of the Ambient-Assisted Living (AAL) and the Enhanced Living Environment (ELE)

The Ambient Assisted Living (AAL) can be considered a subfield of a broader field of Ambient Intelligence (AmI) [1]. It focuses on using ambient intelligence techniques to improve the quality of life and independence of older people. It is a cross domain-initiative that utilizes ICT technologies to best support, primarily older people, in their everyday activities.

The technology advancement and proliferation fueled by the increased demand and cost-reduction trend Internet-of-Things (IoT) technologies are becoming increasingly more intertwined into the fabric of our everyday life. The need for AAL solutions is also partly fueled by the desire of the baby-boomers to keep living in their homes. The AAL can also be considered a segment of the Smart Home market even though still a very small fraction of it. The ICT-enabled independence of older adults can be, in many ways, achieved by building Enhanced Living Environments (ELEs), which are sensitive, adaptive, and responsive to human needs and habits. The ELE is a term coined to refer to where AAL meets and makes use of ICT to sit at the intersection between AAL and AmI, technology-applying ICT for everyday life, and personal environment management [2].

### 2.2. Survey on Anomaly Detection Techniques

To explore the application of the various approaches for activity recognition and anomaly detection, a literature review has been performed, and this section briefly summarizes the results.

Definition of the anomaly detection in ambient environments is an elusive one since it can, in different contexts, mean different things to different people. The anomaly detection, or abnormal activity detection, aims to identify unusual behavior in the environment. It does this in a dynamic way and, with time and appropriate algorithms, it can be trained to increase accuracy. However, in most cases, it can only find symptoms instead of root causes and does not identify issues that do not show (any or correct) symptoms. Anomalies or patterns in data that do not conform to the expected behavior are often referred to as outliers, exceptions, surprises, errors, novelties, contaminants, and similar phrasing, depending on the domain and context.

One of the biggest challenges in anomaly detection is the definition of the “normal” behavior and the boundary that is used for identifying an anomalous observation. Furthermore, the notion of “normal” is not static and can change over time and context. The tolerance for the error is very different in different application domains (such as medical, fraud detection, cybersecurity) and has different implications. Quality and availability of the training data, together with an inherent bias, influences the anomaly detection process to a large extent.

When dealing with anomaly detection in different domains, different approaches are used. However, in general, they include (some or all of) the following: (heterogeneous multi-dimensional) data gathering, information fusion, feature extraction, pattern analysis and recognition, context analysis, and anomaly detection. After the anomaly has been found, the action that was previously defined is executed. Having the ICT-enabled smart homes equipped with sensors enables the data gathering. This data, when cleaned, filtered, and aggregated, is often used to infer higher-level context events that are, in turn, used for additional reasoning. The recognition of users’ daily activities is one possible usage of the data arriving from the smart home and can be applied in different scenarios ranging from safety and healthcare to lifestyle and energy management. Having the answers to the questions of “who,” “where,” “when,” and “what,” the system (lately increasingly using machine learning) can be used for pattern recognition and possibly (depending on the purpose of a system) anomaly detection.

Anomalies can be identified at different temporal scales such as single events, days, weeks, or months. The seasonality aspect is also sometimes relevant, e.g., some activities are characteristic for summer or winter periods only. Some approaches build on the rare occurrence [3] and some on expected temporal relationships between events [4].

When discussing anomaly detection regarding a person living in a smart home, behavior, activity, or pattern recognition are often mentioned. For an anomaly detection system to work correctly, first the model of a normal behavior has to be developed so that deviations from that model can later be identified as an anomaly. Bourobou et al. [5] suggest a combination of a pattern clustering method (based on the K-pattern clustering algorithm [6]) and an activity decision algorithm (based on the Artificial Neural Network algorithm).

Activity recognition systems typically use a sensor-based approach (including Body Sensor Network or Smart-home Sensor Network or both). Based on the types of sensor data, the vision-based and non-vision based (wearables or object-attached) sensors can be distinguished. Based on the way the data is analyzed, the data-driven and knowledge-driven activity recognition can be distinguished (more on this can be found in work by Chen et al. [7]).

According to Meng at al. [8], existing studies on (daily) activity recognition typically follow one of two approaches, i.e., the rule-based approach (relying on manually created rules that are not easy to scale up) and the pattern recognition approach (relying on the machine learning techniques, typically classification methods for pattern recognition). The same authors argue that recognition of Activities of Daily Living (ADLs) and abnormal behavior detection is often limited by an offline analysis strategy, complex parameter tuning, obtrusive data collection, and a need for training data. For this reason, they present Online Daily Habit Modeling and the Anomaly Detection (ODHMAD) model, for the real-time personalized ADL recognition, habit modeling, and anomaly detection for the solitary elders.

Rashidi et al. [9] used motion sensors and interaction tracking sensors to obtain quantitative information about activity patterns. The measured patterns were clustered into different activities, which were later used to develop a Hidden Markov Model (HMM)-based algorithm that demonstrated high accuracy and sensitivity. The work also gives an overview of activity recognition methods ranging from supervised and straightforward such as Naïve Bayes, decision trees, and Markov models to unsupervised and more complex methods.

Ni et al. [10] give an extensive overview of the work on anomalous situation detection and identify techniques such as rule-based approaches, temporal relation discovery, similarity-based approaches, erroneous-plan recognition (for detecting faults in the daily activities of older adults with dementia), near-field imaging, far-field audio, and more. They also give a conceptual description of an activity characterization, activity taxonomy, and activity context representation formalization.

Bakar et al. [11] conducted a survey on activity and anomaly detection in the smart home and concluded that anomalous activities recognition is still immature in the smart home when compared to other domains such as computer security, a manufacturing defect detection, medical image processing, and more.

Chandola et al. [12] provide structure and comprehensive overview of the research on anomaly detection covering several techniques (classification based, clustering based, nearest neighbor based, statistical, information theoretic, spectral) and application domains (cyber-intrusion detection, fraud detection, medical anomaly detection, industrial damage detection, textual anomaly detection, and sensor networks). They identify different aspects of the problem as nature of input data, type of anomaly (point, contextual, collective), data labels availability (where supervised, semi-supervised, and unsupervised techniques are distinguished), and output (scores or labels). Typical anomaly detecting techniques used for anomaly detection in sensor networks are rule-based systems, Bayesian networks, parametric statistical modeling, nearest neighbor-based techniques, and spectral techniques.

### 2.3. Survey on Activity and Anomaly Detection AAL Systems

There are many overviews and extensive surveys of smart homes for senior care such as the one by Majumder et al. [13] where a comprehensive review on the state-of-the-art research and development in the smart home-based remote healthcare technologies is presented. The authors also give an extensive survey and overview of smart home systems prototypes and commercial systems for remote senior care. They classify smart home monitoring systems for seniors and people with disabilities into the following categories: automated emergency call systems, automated activity, fall detection systems, vital signs monitoring systems, reminding systems, and automated health assessments.

Sanchez at al. [14] describe smart home developments and challenges in the last 16 years together with the most used analysis methods for human activity and behavior recognition and their implementations. They identify most implemented analysis methods in the period from 2000 to 2016, which includes the Bayesian network, Naïve Bayes/Decision Trees, Decision Trees, Computer vision, Intelligent Agents (IA), Computer vision/IA, Correlated pattern, Gaussian Distribution, Kernel Density Estimation, Mix models, Maximum likelihood, T-Patter, Hierarchical Classifiers, Quadratic Discriminatory Classifier, Principal Component Analysis/K-Nearest Neighbors, Fuzzy Logic, Fuzzy Logic/IA, IA/Bayesian Net, IA/Lezi, Lezi, Markov model/IA, Markov Model, Neural Network (NN), Markov Model/NN, NN/Support Vector Machines (SVM), SVM, and SV Data Description.

Adam et al. [15] give a review of the assistive smart home projects and AAL technologies together with ambient sensors, typical wearable and mobile sensors, (mobile, ambient, and vision-based) activity recognition techniques, and algorithms.

Ni et al. [10] give an overview of the most significant smart home projects that aim to enhance assisted living for older people. They show how, in the early years, smart home projects focused on physical and logical connectivity of devices, how they have evolved to a multidisciplinary approach focused on improving the usability, and how they have become the enabling technology to achieve other purposes, such as independent living of older persons.

Holmes [16] is an anomaly detection system for daily in-home activities that run for six months in a single student resident home. Activity labels include sleeping, meal preparation, cooking, breakfast, lunch, dinner, snack, dishwashing, watching TV, using a laptop, going to the toilet, showering, and leaving and entering the home. The system, however, starts with an initial set of pre-defined rules and newly detected anomalies have to be verified by the resident to be included as new rules if appropriate.

Suryadevara et al. [17] developed a system for monitoring and evaluating essential daily activities and tested the system at the homes of four different older adults living alone. The system included a wireless (Zigbee) sensor network (detecting usage of electrical devices, bed and chairs usage via pressure, contact or electrical sensors) and identified activities like sleeping, preparing a meal, dining, watching TV, toileting based on the Sensor-ID, and time of day.

Novák et al. [18] presented a service for anomaly detection in users’ activities using data from unobtrusive sensor and alarm issuing if unusual activity is detected (long periods of inactivity, lack of activity, unusual presence, and changes in daily activity patterns). Anomaly detection is based on a composition of unsupervised classification technique Self-Organizing Maps and the next activity prediction using the Markov model.

Lotfi et al. [19] presented a solution for supporting independent living of the elders who have dementia by equipping their home with a simple sensor network to monitor their behavior. Recurrent neural networks were used to predict the future values of the activities for each sensor, and the caregiver was informed in case anomalous behavior was predicted in the near future.

## 3. The SmartHabits System

### 3.1. Overall System Description

The SmartHabits system can be described as an intelligent service that reassures family members that their loved ones are fine. To perform real user evaluations, a complete end-to-end system has been realized using commercially available (off-the-shelf) sensors integrated with a home gateway component transferring sensor data to the cloud-backend services.

The system utilizes Artificial Intelligence concepts of reasoning, pattern detection, decision making, and relies upon advances in Ambient Intelligence (AmI), sensor networks, and Human-Computer Interaction (HCI). It uses non-vision-based sensor data and data-driven pattern recognition (based on probabilistic reasoning and machine learning). By using affordable, unobtrusive sensors in the home, the system learns about person’s typical behavioral patterns (recurrent way of acting) without invading their privacy and alerts her family member(s) or (informal) caregiver(s) if an anomalous situation is detected.

The system does not focus on health monitoring nor life-threatening emergencies but rather on pattern recognition and anomaly detection providing better care assistance, reassurance, and peace of mind. Figure 1 shows an overview of the system. Simple sensors that do not invade the person’s privacy (i.e., no cameras or microphones) send data to the SmartHabits Cloud Platform where it is collected, stored, and used for pattern recognition and anomaly detection by the SmartHabits Expert System. In case an anomaly is detected, an alert is sent to designated family members and/or (informal) caregivers.

### 3.2. The SmartHabits Expert System

The Expert system, as one of the largest areas of application of Artificial Intelligence, emulates the decision-making ability of the human expert and represents knowledge as data or rules [20]. With the advancement of AI, they are also gaining popularity. Expert systems are typically very domain-specific and fall into the following categories [21]: interpreting and identifying, predicting, diagnosing, designing, planning, monitoring, debugging and testing, instructing and training, and controlling.

According to the categorization above, the SmartHabits Expert System is falling into the expert system categories of both interpreting, identifying, and monitoring expert systems and is composed of the following components (Figure 2).
The *Knowledge base*—stores specialized knowledge in the form of facts (e.g., about the user and her environment) and rules (e.g., rules defining which situation will trigger the issuance of notification).The *Patterns recognition engine*—makes inferences about patterns and related rules (e.g., what are the typical movement patterns) based on the knowledge stored in the *Knowledge base*. It delivers new facts based on the existing ones. In the process of deducing the new learnings, machine learning techniques and unsupervised machine learning algorithms are used to process contextual clues and other relevant data (such as system configuration setup and configuration data). For identifying “normal data instances,” the clustering-based technique was used following the assumption that *normal data instances lie close to its cluster centroid while anomalous instances lie far away.* The pattern recognition was done repeatedly as the new data arrived to incorporate the newest information and was done so using a dynamic sliding-window approach [22] to give less weight to the history data and allow the seasonality of the behavior to have more impact.The *Anomaly detection engine*—makes decisions, based on the knowledge stored in the *Knowledge base*, whether some situation can be considered anomalous or not. If an anomalous situation is detected, information about the circumstances leading to that decision (e.g., concerning the condition and related rule leading to that decision) is sent to the *Notification management engine* and to the user. In the detection process, the rule-based forward chaining technique was used (involving checking the condition part of the rule to determine if the condition is true or false).The *Notification management engine*—adapts system messages to make them user-readable (e.g., performs message compositions and language internationalization) and sends it to the *UI* component.The *UI*—is used to exchange information with the user (e.g., for delivering notifications, receiving feedback, and enabling the user to modify rules).

In the anomaly detection process, an unsupervised anomaly detection technique was used because it does not require (labeled) training data. Anomalies of interest fall into the category of *contextual* (a.k.a. *conditional*) *anomalies* since the data instance (in our case, time-series data) is considered anomalous only in a specific context and for its local neighborhood. The output of the anomaly detection falls into the *label type* since the techniques used to assign a label (normal or anomalous) to each data instance.

### 3.3. System Requirements

In the planning phase, the requirements were defined to capture both functional and non-functional expectations of the system and to drive and focus on the development process. Some of them are shown in Table 1.

#### User—System Interaction

Innovative human-machine interaction paradigms built upon the pervasive and unobtrusive communications concepts make the Ambient Intelligence technology a suitable candidate for the realization of Enhanced Living Environments. With the integration of technology in the background, the interaction becomes truly ubiquitous, which makes the concept of Human-Environment Interaction a reality. Exploitation of different technology and contextual information to deliver personalized (tailored to the needs of the individual), adaptive (to the changing needs of the individual), and natural interaction experiences has many challenges.

Today’s applications are becoming increasingly sophisticated in terms of available options and functionality offered. The fact that many parameters are affecting the user interaction experience and expectations (see Figure 3) makes the whole process even more complicated. Thus, it is of pivotal importance to hide this complexity from the user (e.g., by incorporating automation and anticipative mechanisms).

The user interfaces should primarily be useful and usable. They should not overwhelm the users and should be highly intuitive and natural. There are many specific challenges [23] of developing compelling user interfaces for AAL applications. This ranges from the interaction design, adaptivity aspects [24], changing and diverse competencies of the end-users, multimodality support, and trust issues.

In the SmartHabits system, the interaction with the home (older) users was selected to be implicit (see Section 4.1.1.). The interaction with the caregivers was designed to be more granular (e.g., they could receive the notifications, automatically issued by the system, via different communication channels).

### 3.4. System Architecture

The SmartHabits system is an end-to-end system composing of a part that resides in the (Smart) Home Environment (that reads and sends sensor data to the Cloud Platform), the cloud-backend (that collects and analyzes the data and uses it for reasoning and acting), and client applications (used by the informal caregivers and system administrators).

The system architecture is shown in Figure 4 and is composed of the two main parts. The Home-Sensing Platform is envisioned as a logical environment, deployed over a physical space, which is mainly responsible for implicit interaction with the home user through sensors. The Cloud Platform acts as a central processing and data storage system and can support any number of (Smart) Home-Sensing environments.

#### 3.4.1. The Home-Sensing Platform

Home-Sensing Platform components residing in a local home user’s (Smart) Home Environment are:The *Device Gateway* (DGW)—responsible for integration and configuration of the different sensors and relaying measurements to the respective processing components via the Local Database.The *Home Gateway* (HGW)—responsible for data processing, data storage, configuration, and connection with the Cloud Platform services. Data persistence (using Local Database) and buffering in case of the (Internet) connection loss with the Cloud Platform are among the main functionalities to enable data preservation.The *Local Database* (LDB)—For data persistence, the document-oriented NoSQL database Apache CouchDB [25] was used.

#### 3.4.2. The Cloud Platform

The SmartHabits Cloud Platform is based on microservices architecture providing great flexibility for responding to the needs of different scenarios. The cloud services expose their functionalities primarily via REST (REpresentational State Transfer) APIs using lightweight JSON (JavaScript Object Notation) as a data interchange format. All APIs are secured by JWT (JSON Web Tokens). Apart from using (operating-system and programming language agnostic) APIs, the AMQP (Advanced Message Queuing Protocol) is also used for the communication.

The Cloud Platform components that can be deployed on single or more machines and run on physical or virtual machines on any cloud operating system are:The *Cloud Gateway* (CGW)—responsible for receiving processed and indexed measured data from HGW, storing the data, and (re)configuration of different (Smart) Home Environments.The *Event Reasoners* (ER)—responsible for inferring higher level *ContextEvents* based on the: a) *SensorEvents* arriving from the Home environment(s), b) most current information about the Home-Sensing Platform setup and configuration, and c) information about the related home user. The component is subscribed to the *SensorEventBus* and publishes *ContextEvents* on the *ContextEventBus* utilizing the messaging pattern.The *Profiling Server* (PS)—responsible for managing user-related data including user profile information.The *Notification Manager* (NM)—responsible for managing (e.g., constructing, storing, forwarding) all notifications and alarms to specific users based on set preferences and priorities. The notifications are forwarded to and delivered: a) as notifications in a native mobile application (Mobile Client), b) as SMS messages (utilizing SMS gateway), c) as email messages, and d) as notifications in the Web portal (Caregiver Portal).The *Login Service* (LS)—responsible for handling authentication and role-based access to the system.The *Rule Recommendation Engine* (RRE)—responsible for learning user behavioral and environmental patterns and proposing related rules. It utilizes machine-learning techniques to process the data and provide new insights. It also performs redundancy checks by utilizing the knowledge of the “similarity” to assess the rules of similarity measures and provide the optimal set of the rules to be presented to the caregiver user.The *Rule-Based Anomaly Detector* (RBAD)—responsible for managing rules, detecting anomalies based on (manually and automatically) defined rules and, when appropriate, notification triggering.The *System Portal* (SP)—a web-based portal for System Administrators providing interface and support for administrative and technical management of the system. Interacts with cloud-backend services exclusively via the System Portal Backend (SPB) component acting as an API Gateway (a.k.a Backend for Frontend, one of the microservices design patterns for effective collaboration offering a single-entry point for all interactions).The *Caregiver Portal* (CP)—a web-based portal for Caregivers. Interacts with cloud-backend services exclusively via the Portal Backend (PB) component acting as an API Gateway.The *Mobile Client* (MC)—a mobile Android-native application with the lightweight functionality of the CP.All components in the system use a common *data model*, which is based on the well-established ontologies such as IEEE Suggested Upper Merged Ontology (SUMO), OntoSensor ontology, Semantic Sensor Network (SSN) ontology, and CSIRO Sensor Ontology. In the process of developing a data model, the guideline from the ETSI Study on Semantic support for M2M data, which says that any piece of information in the information model should be expressed in a unique way, was followed.Depending on the purpose of all Cloud Platform services, use of isolated data storage for each component (microservice) as either relational (MySQL [26]) or non-relational (MongoDB [27]) database or both. For the publish-subscribe functionality, the open source message broker software (sometimes called the message-oriented middleware) RabbitMQ [28] was chosen.

Considering the SmartHabits Expert system functionality, the *Patterns recognition engine* is mostly implemented within the RRE component, the *Anomaly detection engine* functionality is mostly contained within the RBAD component, the *Notification management engine* is implemented by the NM component, the *UI* component is implemented by the Mobile Client, the Caregiver Portal, and the System Portal, and is facilitated by the SMS and email gateways. One of the most critical functionalities delivered by the *Expert System* is anomaly detection (explained in 3.5).

#### 3.4.3. The Caregiver Web Portal

The main idea of the system was that it is adjustable to end-user preferences especially concerning privacy and what information the personal data home user is sharing with his caregivers or family members. In some cases, the caregiver will not have access to sensor data but will only receive notification when something unusual happens. Since different people have different needs and preferences, several ways (channels) of communication were made available for caregivers. Those channels include SMS, Mobile (Android) Client, e-mail, and the Web Portal where the latter allows for the highest number of functionalities and more detailed and explicit interaction.

Via the Web Portal, the caregivers can, depending on the preference, change different aspects and configuration of the system, see an overview of home status, manage learned patterns, and even manually add their own rules. Figure 5 shows the Home Status view where the current home status overview can be seen. On the left pane, the information about the currently selected home, the home online/offline status, the number of rooms where the sensors are placed, and the number and type of installed sensors that can be seen. Alongside the first pane, every defined room is listed (right and below) with related sensors and their last readings. At the bottom of the screen, the user can access other views such as:*Rule Management*: for optional adding, editing, or enabling/disabling three types of rules: appliance rules (related to power sensors and plugged in *appliances*), status detection rules (related to motion and reed sensors), and environmental rules (related to temperature, illuminance, and humidity sensors).*Notification History*: showing all notification history related to the selected user containing the notification type (info, warning, alert), message text, receiving time.*User Profile*: showing the details about the selected user that can be modified such as username, gender, first name, last name, address, city, country, birthplace, birth date, and mobile number. All details, except username, are optional and are not crucial for system functionality. This view also allows us to enable and disable the system learning (and rule suggesting) capability.*Home Configuration*: showing all appliances (connected to power sensors) and sensors installed in the home of the selected user together with their name, type, and indoor location (room).

Figure 6 shows the details of one selected rule. On the top, the details of the rule can be seen (and modified). In this case, the rule can be translated as “*Notify me if the motion in the living room is not detected in the period between 19:11 and 23:56 on any day in the week.*” Below the rule details, the related sensor data (coming from the motion sensor) can be seen for the period of one month. Each row represents one day from 0 to 24 h, and the legend explains the colors where darker colors mean more movement in the 10-min intervals. The vertical (red) lines represent the learned and suggested boundaries for the rule (19:11 and 23:56).

### 3.5. Anomaly Detection Process

For the anomaly detection process to be successful, certain preconditions had to be met. When the right task for the project was defined and the underlying infrastructure was set up, the process of preparing data for the machine learning algorithm, and incremental model building, was conducted iteratively. The context for which the SmartHabits was developed included behavior anomaly detection for the independent living of older people. The goal was to identify unusual behavior in the self-living environment (usually used living space for independent living such as a flat or a house). Starting pre-conditions included recognition of the usual behavior of a monitored person in their living environment by using non-obtrusive, non-wearable, privacy-aware, and cost-effective data gathering. Data collection included multi-dimensional data gathering for living pattern recognition in a time scale of a day, week, or month. Based on recognized living patterns, the selection of most usual (repeating) pattern and characterization of a personal profile in a unified way were done. Some examples are spacing curtains every morning between 7:00 and 7:30 am, cooking coffee or tea every morning from 8:30 to 9:00 am, watching the TV at 9 am or a similar activity that can be recognized normal. The goal of the pattern recognition was to derive the boundaries of the rule to be presented to the (caregiver) user. If the rule is accepted (manually or automatically, depending on the user preferences) and the conditions of triggering the action are met, the action is executed.

The anomaly detection process is comprised of four steps: data collection, data pre-processing, data analysis, and anomalies identification. Data collection was enabled by the Home-Sensing Platform together with two of the three data pre-processing steps, which is a data format and cleaning, while the sampling was done in the Cloud Platform (in RRE). For the data format, the open-standard JSON (JavaScript Object Notation) syntax for storing and exchanging data was selected, using human-readable text consisting of attribute-value pairs. The cleaning process was done within the HGW component and included validation of both *SystemEvents* (containing information such as the sensor battery level, the sensor error, and the time-stamp error) and their specific payloads (values and types of different measurements). The data sampling process was done in the Cloud Platform on a regular basis (once per day at 3 M). Previously, the data (in the form of *ContextEvents)* was fed to the RRE for ER (as explained in Section 3.4.2) sampled the data transformation and aggregation. Data in focus is the unlabeled time-series data, a collection of the time-stamped entries (*ContextEvents*) ordered in time, and fed to the unsupervised learning algorithm. Regarding the anomaly detection technique, the clustering-based method was used. The advantages are the low time complexity and broad application since it can be used for several data types.

Feature selection, extraction, and construction were done to identify relevant data and improve the quality of the data before it is presented to the learning algorithm. Both categorical and continuous classes of features were used. Based on the strategy of searching the embedded feature, a selection method was used in which the feature selection was performed during the model construction process. User behavior or the environment pattern data model contained information such as a home user id, a unique pattern id, a sensor id, a derived measurement dimension, a variable value mean (mean of the observed dimension obtained during the model training period), a variable value standard deviation, a variable value lower boundary, a variable value upper boundary, and a rule id (unique id of the rule pertaining to the pattern).

Defined *ContextEvent* dimensions were time switched on, time switched off, duration, duration per day, duration per period, movement detection time, time reed opened, time reed closed, change rate, and value. Selected features correspond to the defined (*ContextEvent*) dimensions. For dimensions where it made sense, *k-means* clustering method (R implementation that guarantees optimal and reproducible results) was used to identify natural clusters (partitions) in the data. For the duration and change rate dimensions, the clustering was not completed. The number of clusters (*k*) was defined automatically by using the algorithm that yields an optimal number of clusters using the BIC (Bayesian Information Criterion) measure. In the process, the following configuration parameters were included: size of the dynamic sliding-window (30 days), a minimal cluster size (10), minimal sample size (7 days), sampling period (1 min), a maximum number of clusters (3), a rule similarity threshold, and similar factors. The algorithm, realized within the RRE component, was used to find groups of unlabeled data related to the user behavior and environment. For each 1D (one dimensional) rule, the average day distribution was found for each observed dimension. Before calculating pattern parameters (boundaries) by bootstrapping confidence intervals, the distinct patterns were found in cases where multiple patterns were possible.

Found patterns were compared with existing patterns to identify both possible small and negligible differences in value boundaries and similar distribution (Wilcoxon or Mann-Whitney test). In a case where a similar pattern did not exist, the new rule was suggested. In case there was a similar pattern already in the database, the pattern was updated, and, if there was a rule bound to the pattern, its suggestion was resubmitted in case its state was either “deleted” or “rejected” and the time to resubmit (configurable parameter, e.g., 30 days) elapsed.

Figure 7 shows the basic rule-based anomaly detection data flow. Based on the rule definition, RBAD is either subscribed to specific context event topics or scheduled for monitoring a specific context situation. If the searched situation is detected, a notification is triggered. This shows basic rule-based anomaly detection data flow. Based on a rule definition, RBAD is either subscribed to specific context event topics or scheduled for monitoring a specific context situation. If the searched situation is detected, a notification is triggered.

## 4. Pilot Case

To evaluate the SmartHabits system in the real environment, a pilot was set up in the city of Zagreb in cooperation with the Foundation taking care of the older people living alone. Before the start of the pilot, all legal requirements were met, and specific procedures were followed. After the preconditions were met and the end user recruitment was done, the on-site installations took place. Overall, the pilot had run for six months after which the feedback from all involved users was collected and analyzed. In this section, we share details on the pilot case and setup.

The goal of the pilot was to see if the trial ready prototype of the SmartHabits system could successfully learn typical daily patterns, detect unusual situations in the household of the older person living alone, and notify his or her caregiver when an unusual situation is detected. Special emphasis was put on the assessment of the human-computer interaction aspect and the evaluation of the usability and efficiency of the system.

### 4.1. Targeted End-User Groups

Overall, 13 different users participated in the pilot including home users (older people living alone), caregivers (each with granted access to only the designated home user), and administrators (responsible for the system and pilot setup). The number of the Home-Sensing Platforms, and, thus, home users, was hugely influenced by the cost of the hardware needed for the installation and less by the effort needed for the (software and hardware) installation and support.

#### 4.1.1. Home Users

Home users in the scenario selected for the pilot were older people living alone belonging to a particular risk group, but, otherwise, in good health. According to the Canadian Study of Health and Aging (CSHA) 7-point Clinical Frailty Scale [29], the home users selected for the pilot belonged to the 4^th^ category (*Vulnerable—while not dependent on others for daily help, often symptoms limit activities. A common complaint being “slowed down” and/or being tired during the day*).

Home users did not have to interact with the system explicitly, which allowed them (to a large extent) to forget about the system. The idea was that they continue with their usual daily activities with no need to perform any additional tasks. Not needing to interact with the system explicitly or immediately made the system more suitable for the broader user group (since, e.g., there was no need to design, develop, and explicitly tailor the UI to address specific needs, possible impairments, and preferences of the home users). Another important design decision, coming from the discussion with the end users, was to avoid using any Body Area Network (BAN) technology that the users should wear or remember to charge and put on (on their body [30,31] or their clothing [32]).

The value proposition for home users is that they can enjoy better independence with less stress (e.g., no one will know if something unwanted happens).

#### 4.1.2. Caregivers

Target users, in terms of value delivered by the system, are caregivers, who are typically family members or paid helpers, who regularly take care of the older adults (home users) that might need assistance and reassurance. Unlike home users, target users interact with the system explicitly. They receive notifications from the system if something unusual has happened in the user’s household. Some examples of warning messages (being a result of the observed pattern) are:“There was no motion detected in the living room until 8:10 am.”,“Front door was opened for more than 15 min.”,“The oven was on for more than 3 h.”,“The coffee machine was not turned on until 11 am.”.

Caregivers can also access the Web Portal that enables them to see an overview of home status, to manage learned patterns, and even to manually add their own rules for notifications if needed. It is important to emphasize that only the caregivers who were previously approved by the home users can gain access to the SmartHabits system.

The value proposition for caregivers is that they can have peace of mind knowing the system will notify them if some unusual situation is detected in the home of their loved one. As an additional offering, they can access the web-based dashboard showing the home status and home user’s activity at a glance and, if wanting, they can see detailed information from every sensor and add or customize recommended rules and notification preferences.

#### 4.1.3. Administrators

Administrators acted as service providers performing administrative tasks related to the SmartHabits system, such as system provisioning. Utilizing their technical skills, they were responsible for the Home-Sensing Platform installation and configuration, hardware deployment, and system health monitoring, and acted as the first line of support to other (categories of) users.

### 4.2. Legal and Privacy Considerations

There are many legal and privacy challenges when developing an AAL system. One of the examples is the study “*Legal Aspects on Smart House Welfare Technology for Older People in Norway*” [33] where the following main legal challenges were identified: data privacy, data access and management, stakeholders’ interest, and informed consent of the users and/or the users’ families.

The SmartHabits users (both home and caregivers) gave written consent confirming they agree to use a system and understand how the data collected will be used. Before signing such a consent, a training workshop was organized where the system was presented together with the pilot goal, type, and purpose of the data collection. Before the workshop with the older users, we have identified potential questions with the experts from the foundation (taking care of the older people living alone) and formed the agenda accordingly. Additional presentation of the system and data management procedures was additionally done at the time of the Home-Sensing Platform installation to make sure everything was clear to our users.

Following the EU General Data Protection Regulation (GDPR) practices, data collection related to the run of the pilot was registered with the Croatian Personal Data Protection Agency.

### 4.3. Hardware Used in the Pilot

One of the main challenges was to keep the cost of development, deployment, and running as low as possible to cost-effectively facilitate the use case and test the concept considering business and scalability aspects. As mentioned in the introduction, the big challenge of AAL systems is delivering benefits for the end users in a cost-efficient and sustainable way so that the business value is higher of the overall delivery costs. For that reason, the price of the hardware, apart from the other resources, was kept as low as possible.

In the hardware selection process analysis of the Single Board Computers (SBCs), a low-cost and power optimized compact single-board based computer was done. The reason for opting for SBC as the hardware for installation and running of SmartHabits Home-Sensing Platform was the availability of all functional components of a computer like memory, I/O capabilities, and a microprocessor. Analysis framework and survey details are beyond the scope of this paper. However, some more important analysis parameters were: attractive price, low power consumption, multi-tasking capability, Wi-Fi module availability, extension capabilities, and widespread adoption.

Considering the results of the analysis, the Raspberry Pi (RPi) 3 was chosen. As a third-generation Raspberry Pi, version 3 was released in February 2016 with some specifications being: Quad Core 1.2GHz Broadcom BCM2837 64bit CPU, 1GB RAM, onboard Wi-Fi, 4 USB 2 ports, full-size HDMI, Micro SD port for loading the operating system, and storing data.

For detecting and responding to inputs from the physical environment, an additional analysis of sensors was made. Some parameters driving the sensor analysis and the selection process were: sensor packaging, reliability (performing the intended function in nominal conditions), robustness (ability to withstand adverse conditions), accuracy, price, size, battery support, configuration support, ease of installation, market availability, and an energy-efficient communication protocol. Again, a literature survey was performed that included examining and comparing the competitive technologies (e.g., review of smart home technologies by Lobaccaro et al. [34] where pros and cons of different technologies are presented).

Considering the results of that analysis, the following off-the-shelf sensors were chosen: Fibaro Door/Window Sensor (as a door and/or window sensors), Fibaro wall plug (as a power sensor), and Aeotec Multisensor (as motion, light, temperature, humidity and UV sensor).

All selected sensors (shown in Table 2) operate on the temperature 0–40° and use a Z-Wave communication protocol designed to specifically control home appliances and sensors. The protocol offers reliable and secure communication, low-power consumption, and simple installation with remote and local control of connected devices.

Apart from the information reflected in the table (Table 2), the specification of the Multisensor 6 includes the following information: measured temperature range: −10 °C to 50 °C, accuracy: ±1 °C, measured humidity range: 20%RH to 90%RH, accuracy: ±3%RH (at 25 °C), lighting: 0 LUX to 30,000 LUX, and max motion sensitivity: 5 m.

For the Z-Wave protocol support (the protocol used by all our selected sensors), several RPi extensions were trialed in our Lab, and, in the end, the RaZberry board proved the most appropriate. One of the biggest reasons being that it fits the RPi case, which made the on-site installation much easier and more robust (since e.g., Z-Wave USB dongles were easier to disconnect by accident). The ZWay Server was used as a device gateway and provided the admin password protected account and a portal for (re)configuring device parameters (such as wake up intervals, sensitivity, the frequency of reporting, etc.). Home-Sensing Platform was additionally secured by changing the root password, setting up the firewall blocking everything except SSH access to RaspberryPi, HTTP towards Cloud Platform, and NTP (Network Time Protocol). By blocking access to the RPi and additionally securing the ZWay Server access, no new Z-wave device could be added. The sensor devices should support enhanced security function (Security Command Class) to be added securely and utilize encrypted data exchange using AES encryption. RaZberry supports Z-Wave Security S2 and Smart Start technologies. Sensors used in the pilot support a Z-Wave network security mode with AES-128 encryption.

The rationale for using power sensors is that, by observing the usage of household appliances that are regularly used by the home user many activities (like watching TV, listening to the radio, tea/coffee making, meal preparation, etc.), it can be identified. The rationale for using reed (door/window) sensors and movement sensors was the fact that they could provide information for entering/leaving some room or home, using some cabinet, closet, window, or movement inactivity periods, the time of getting up and going to sleep, etc. The rationale behind using environmental sensors was that such information could also provide valuable insight such as if the curtains were moved if the heating/air-condition was on or off and similar. All these activities and situations were successfully recorded and used in the system for pattern recognition, rule recommendation, anomaly detection, and notification triggering and sending.

For Internet connectivity, the mobile broadband Wi-Fi routers were used, partly because of speeding up the on-site deployment and partly because our home users did not have Internet access. In general, the routers do not have fixed Internet connections.

The system is highly extensible to support different devices from different vendors since it abstracts both the notion of a device and sensor. Registry of digital twins, which represent devices and other entities managed by the platform, was enabled. Upon registration of a device, all detailed characteristic information is stored in the system. The measurement payload arriving in the Cloud Platform is agnostic to the data type where it is correctly transformed (in ER) using the system and device configuration data. The design of both the data model and the whole system was made open to extensions to be as future proof based on the existing practices (e.g., from SPOT [35], VITAL [36], and other) and our previous experience in the field.

## 5. Validation Results

The system was successfully validated. It performed its primary function of detecting an unusual situation (an anomaly) and notifying the caregiver(s). During the pilot and testing phase, the system was able to learn, on average, 23 patterns per single household in the first 30 days of the usage (using six sensors listed in Table 2). The average number of learned patterns per different sensor is shown in Figure 8. These patterns were presented to the caregivers in the form of rules, which were either auto-accepted (applied automatically) or proposed (the caregivers decide whether to accept or dismiss the proposed rule).

On average, 61% of proposed rules were accepted by the caregivers, which indicates that the majority of learned patterns were adequate and can be used to offer them reassurance that their loved ones will be fine. The percentage of the accepted rules could be increased by placing some sensors in more meaningful locations such as a temperature sensor in the kitchen, a humidity sensor in the bathroom, and a motion sensor in the living room. However, this claim was not possible to validate in this pilot because environmental and motion sensors were co-located on a single device.

In quantifying the uncertainty of an estimate, confidence intervals (CIs) were used to add bounds on a mean population parameter estimated from the sample of independent observations. For calculating confidence intervals, the bootstrap resampling method was used as a nonparametric method. In our case, the confidence intervals likelihood, or a significance level for the confidence level, was 95% and the lower and upper bounds were periodically calculated for each variable value corresponding to the relevant observed *ContextEvent (CE*) dimension. For the variable value lower boundary, a lower boundary of the confidence interval of the observed dimension obtained during the model-training period was used (taken as a median of the bootstrapped 2.5%-quantile distribution). For the variable value upper boundary, an upper boundary of the confidence interval of the observed dimension obtained during the model-training period was used (taken as a median of the bootstrapped 97.5%-quantile distribution). The number of true positives (TP) in our case was zero since in the period users used the system, the older user was fine, which means that some life-threatening or emergency arose. The situations recognized by the system as anomalies could be interpreted as false positives (FP). Considering this, we calculated the *error rate* (also known as the misclassification rate) representing the information on how often our classifier was wrong and *accuracy* representing the information on how often our classifier was right. For calculating the confidence interval (CI) containing a range of values for a variable of interest including the true value of that variable in 95% of the cases, the Z-score (also known as standard normal deviate or normal score) of 1.96 (corresponding to the 95% probability) was used. The following table (Table 3) shows the mentioned values for one pilot deployment.

It is important to mention that the number of detected anomalies is influenced by the fact that the rule similarity measure was introduced (mentioned in Section 3.5) in order to reduce the number of “too similar rules” to be presented to the caregiver user, which optimizes the set of rules and improves the user experience. The margin of error could be further reduced by increasing the number of samples (*ContextEvents*), which is possible by increasing the time period (in our case, set to 30 days) or by having more active older user (which is not the goal).

At the end of the pilot, all users completed the final survey, which indicated their satisfaction level (as shown in Table 4) and gave additional useful comments. The System Usability Scale (SUS) [37] developed (by the chair of the ISO committee developing ISO 9241-1998 focusing on usability) to quickly measure how people perceived the usability of computer systems primarily inspired the questionnaire.

Some comments from the home users resulting from the free form question *“Is there anything else you would like to share about the system?”* are:-Comment: *The light coming from the hardware was sometimes too distracting.*
○Rationale: During the night, even the small amount of light (in this case emitted by motion and power plug sensors) can be distracting and can bother people, especially when being installed in the center of the home (e.g., alongside the TV). Although a nice feature, it would be nice to have the possibility to turn off those lights or use different sensors of the same type.-Comment: *In addition to the existing functionality, it would be nice to have flood and smoke alarms.*
○Rationale: We have opted not to include these types of sensors in the pilot setup since the more focus was given to the sensors providing data from which relevant patterns were to be inferred. However, such sensors could very easily be added.-Comment: *It would be nice to have a fall-detection functionality as well.*
○Rationale: Some of our home users were especially afraid of not being able to call for help in case of a fall. Since there is much work already invested into fall detection (e.g., sensing-floors, camera-based fall recognition, panic buttons, voice-initiated alarms, and accelerometer-based fall detection) and, since there is no simple, privacy aware, cost-effective, non-wearable, off-the-shelf sensor, we have opted not to include it in the pilot. However, such functionality can be easily integrated with the system in the future.

The home users, in general, liked the system. The fact that it was privacy aware and that it focused on solving the one pain-point helped them understand the benefit of using the SmartHabits system. The decision to use simple, unobtrusive sensors that do not invade the person’s privacy (i.e., no cameras or microphones) helped in both the user recruitment and the system acceptance aspects.

The following table (Table 5) shows the mapping of the system (functional and non-functional) requirements (introduced in Section 3.3.) to implement system components (explained in Section 3.4.) to observe user feedback gathered from the users after the pilot has been finalized. All requirements defined in the planning phase of the project have been satisfied and successfully validated.

For the duration of our pilot, we did not have problems with the autonomy of the sensors, but, for more prolonged and more extensive use, this would undoubtedly be an issue since it would imply an on-site change of the batteries and, thus, disturb the home user. The automatic resets of the Internet connection in the mobile Wi-Fi routers, on which we did not have the control, posed some disturbance in the sense of receiving system-offline notifications but did not otherwise affect the core-system functionality. In this case, the decision to support persistent buffering in case of no Internet connection proved to be very valuable, and no sensor data was lost because of it.

Stability of the RPi as a gateway proved to be satisfying. Some minor bugs and hardware failure (SD card malfunction) appeared but were successfully resolved with no major repercussion on the pilot run.

For more accurate information on the status and activity in the home user’s household, more sensors regarding quantity, (e.g., more movement sensors in different rooms) and additional sensors in terms of type would be necessary. However, too many sensors also pose a problem in terms of installation costs, energy consumption, system-complexity, and privacy. Striking the right balance between number and types of the sensors is a very delicate process and has significant implications on the functioning of the system, its price, accuracy, and robustness, and on the benefit delivered by the system. Achieving the right balance regarding hardware and software setup, economic factors, privacy aspects, and the user interaction experience is not at all trivial, and human cultural and social aspects can affect it.

The user-system interaction aspect was one of the crucial ones to address since even the best functionality if not exposed to the user appropriately will not deliver the wanted impact. For those reasons, we have focused on using rather simple off-the-shelf sensors that do not (in a more considerable extent) invade privacy (like, e.g., microphones or cameras). We also opted for no need for an explicit interaction for home users to not bother them and to make the system “blend into the environment” (following one of the postulates of the Ambient Intelligence). Those design choices made our system much more appealing to the home users and lowered the barrier, which we confirmed in our evaluation discussion with them (before, during, and after the pilot phase).

## 6. Discussion

The acceptance of some systems on the market depends on many factors. Over the years, we had the privilege to be a part of different projects (including EU projects) in the area of AAL where different systems and services were being validated with real users. When designing some service or a product, often immature and too advanced technology is used regardless of its appropriateness for the targeted (end-user) segment, which makes the gap between the offered and the needed system bigger instead of smaller.

Not all users are the same, and the segment of older people is a very heterogeneous one. Some are more technologically savvy and willing to accept technology while others are not. Some perceive the particular system as too obtrusive while others accept it much more easily. The important thing is that the system should meet a specific need and the value it offers should be more significant than the drawback (e.g., the benefit of using it should largely outweigh privacy and trust issues). Unclear benefits with too many functionalities make the system harder to understand and, thus, also unlikely to be used.

The challenges of delivering future AmI and AAL systems should be solved in a holistic way by interdisciplinary teams with broad areas of expertise. A wide-scale inclusion is needed to bring down barriers, from older people and their representative organizations to product designers, researchers, companies, policymakers, and countries. Many aspects affect the acceptance and wide-adoption of systems in this field so stakeholders such as regulatory, standardization, legal, regional, and national bodies should be included in supporting the engineers. Right system design, device selection, right sensor positioning, interaction-design, cost, intelligence-support, ethical issues [38], privacy, security, and safety aspects are only part of the problem to be addressed.

AAL systems have the potential to enhance the quality of life of target users, but we are aware that the set-up of such systems on the market is not trivial and that there are still many challenges to be solved. However, we are confident that the synergetic approach combined with the increased awareness of the benefits and technological advancements will lead to the materialization of that vision.

It is becoming more clear that delivering tightly integrated vertical systems has its benefits but sometimes even greater drawbacks. Open platforms including both hardware and software allow for the specializations, synergies, ecosystems, and added-value creation that would, otherwise, not be possible. Being able to avoid vendor lock-ins and select a best-of-breed offering while building and delivering new and innovative services is, at the same time, lowering costs and focusing scarce resources on creating the most impact, which is very important.

The AAL market can be considered an umbrella market with several segments that are sometimes more and sometimes less directly connected. When one starts to build an AAL product, it becomes apparent that, apart from devices and different hardware, the network and communications need to be available to make some (typically cloud-based) service deliver the benefit as intended. Such AAL systems, thus, have to cut across the chasm between hardware and software, between local and cloud, between less and more computationally and bandwidth heavy, between less and more privacy intrusive, and between affordable and technologically highly advanced (and, thus, more expensive).

## 7. Conclusions

The SmartHabits system was developed and validated in the pilot study to record the feedback of home (older adults) and other users. Apart from validating different parts of the system in the real-world live production setting, additional aspects, such as socio-cultural factors, were investigated as well. The experience gained from the pilot has confirmed that the proposed system can easily transform the typical household to the smart-household, which improves the quality of care. By utilizing simple smart-home sensors that can provide essential and continuous information about the status of the occupant and the environment, we have successfully demonstrated one case scenario focused on prolonging the independence of the older persons living alone while offering peace of mind to their caregivers and/or family members.

The more significant challenges lie in the trust in technology and building the ecosystem, including additional stakeholders and lowering the operational cost to the level where the higher adoption can be expected. Widening potential market applications was also investigated and showed promise. Undoubtedly, future research should include more use cases and possibly with an even more extended pilot period to measure the long-term benefits.

## Figures and Tables

**Figure 1 sensors-19-00907-f001:**
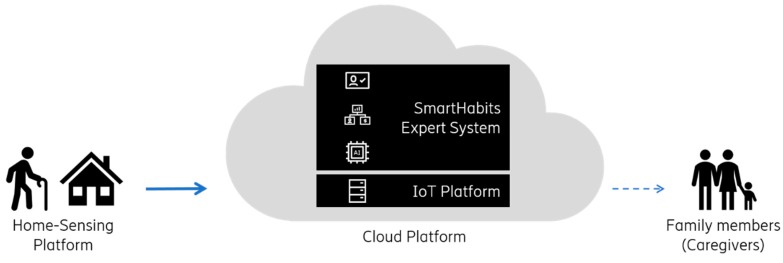
The SmartHabits system overview.

**Figure 2 sensors-19-00907-f002:**
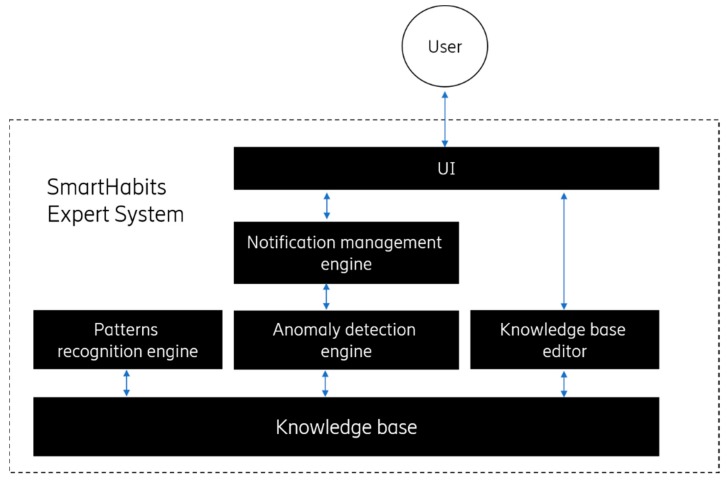
The SmartHabits expert system.

**Figure 3 sensors-19-00907-f003:**
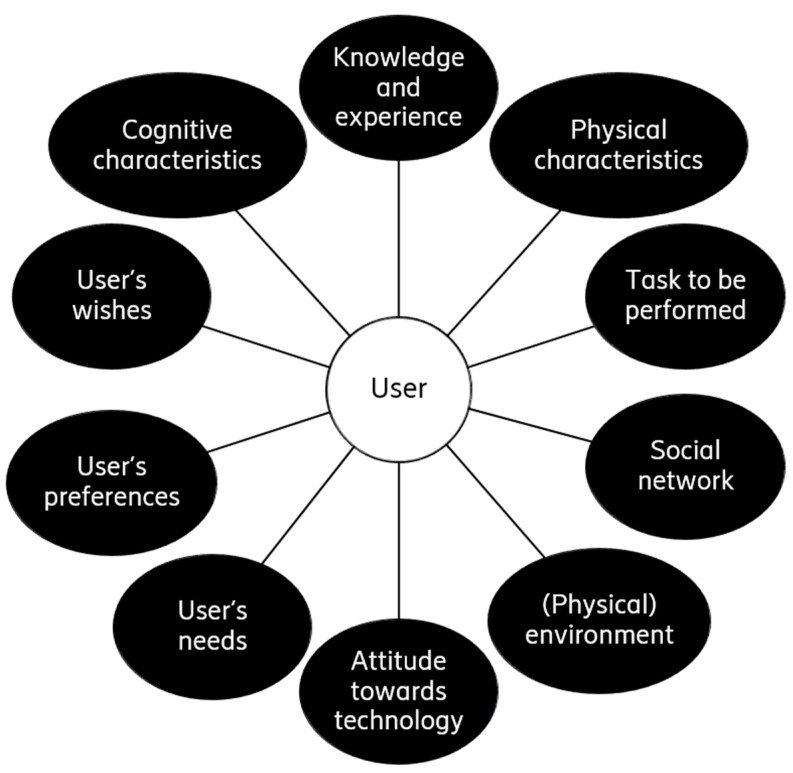
Parameters affecting the user interaction.

**Figure 4 sensors-19-00907-f004:**
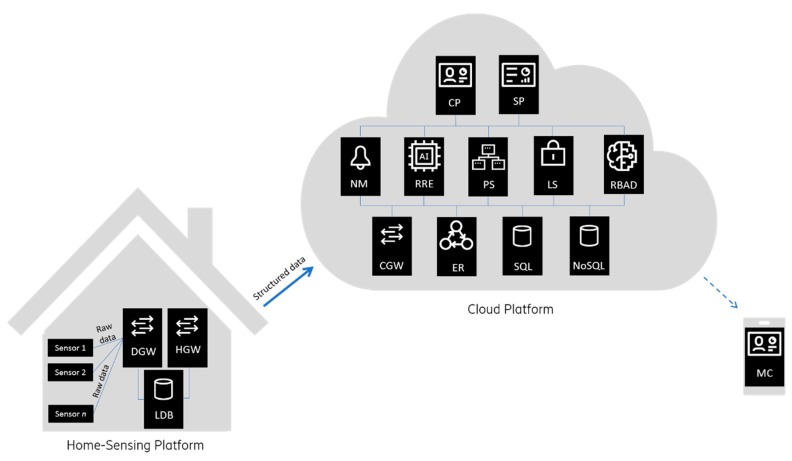
System architecture.

**Figure 5 sensors-19-00907-f005:**
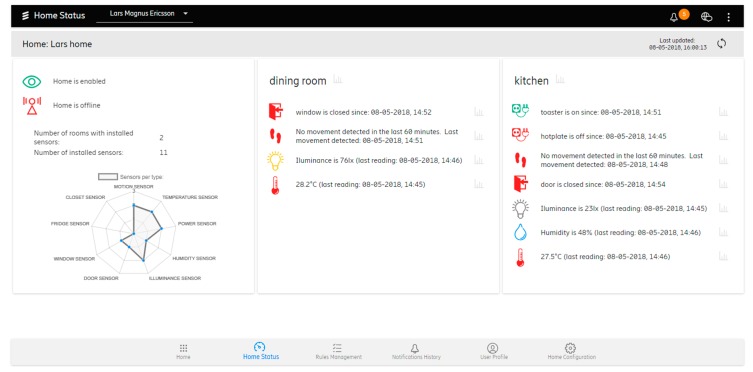
The home status view.

**Figure 6 sensors-19-00907-f006:**
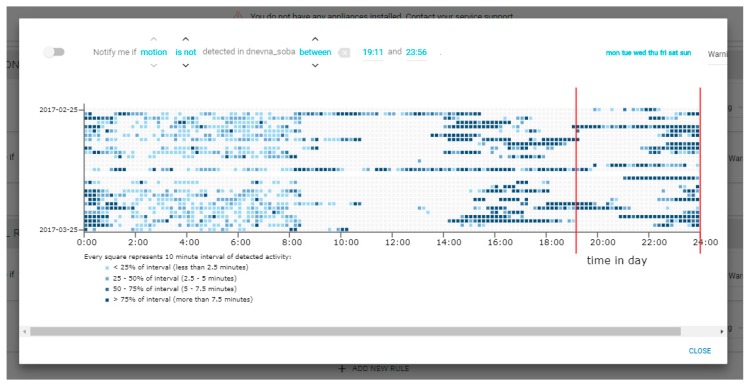
The rule details view.

**Figure 7 sensors-19-00907-f007:**
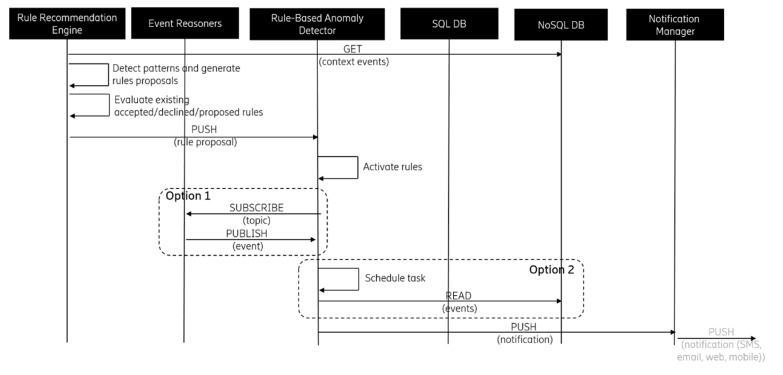
rule-based anomaly detection flow.

**Figure 8 sensors-19-00907-f008:**
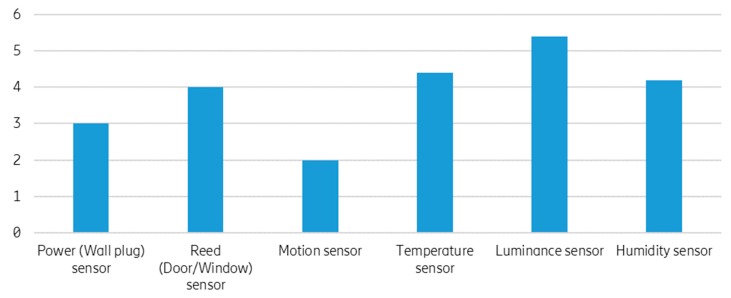
The average number of learned patterns per sensor.

**Table 1 sensors-19-00907-t001:** Selected mandatory system requirements.

Requirement ID	Short Name	Description
Fun_Req_01	Learning support	The system shall learn about the usual user habits.
Fun_Req_02	Rules suggestion	The caregiver shall be able to accept or ignore rules suggested by the machine learning module.
Fun_Req_03	Rules management	The caregiver shall be able to add/edit/delete and activate/deactivate rules for triggering notifications.
Fun_Req_04	User management	The administrator shall be able to add/edit/delete users and enable/disable their access to the system.
Fun_Req_05	Warning notifications	The caregiver shall receive a warning notification (in near real-time) if an anomaly in the user environment is detected.
Fun_Req_06	Home environment status	The system administrator and caregiver user shall be able to see the status of the home system (sensors, home gateway) through a web portal interface.
Fun_Req_07	Home environment management	The system administrator shall be able to provide the data needed for managing the home system (sensor list, room list).
Fun_Req_08	Local buffering	The Home-Sensing Platform shall persist in sensor data locally in case of no connection to the Cloud platform for at least seven days.
Fun_Req_09	Automatic start	The Home-Sensing Platform shall automatically start (i.e., after the hardware restart).
Fun_Req_10	Notifications history	The caregiver and system administrator shall be able to see the history of sent notifications.
Fun_Req_11	System notifications	The caregiver and system admin shall receive warning notification if a home system malfunction is detected (i.e., some sensor has stopped working; there is no connection with the home gateway).
Fun_Req_12	Adaptability	The system shall be able to adapt itself and fit its behavior to changes in the environment and its (re)configuration.
Non_Fun_Req_01	Configuration support	The parameters for (machine) learning module and rules suggestion module shall be configurable.
Non_Fun_Req_02	Non-invasiveness	The system shall be non-invasive and privacy aware.
Non_Fun_Req_03	Zero-touch	The system shall require no explicit interaction from the home user side.
Non_Fun_Req_04	Understandability	The system shall be easy to understand and manage.
Non_Fun_Req_05	Affordability	The system should be as affordable as possible.
Non_Fun_Req_06	Omnichannel notification delivery	The caregiver shall be able to select the preferred way for receiving notifications (email, SMS, Android client app).
Non_Fun_Req_07	Appealing look-and-feel	The system shall have an appealing look-and-feel. The information shall have a nice visual presentation.
Non_Fun_Req_08	Internationalization	The system shall support internationalization (language localization).
Non_Fun_Req_09	Security	Communication between all the main components shall be secured and sent via an encrypted channel. System components shall identify, authenticate, and authorize an entity (human users and other system components) that want to use them before allowing them access to resources.
Non_Fun_Req_10	Scalability	The caregiver shall be able to manage multiple systems (belonging to multiple home users).
Non_Fun_Req_11	Supportability	Caregivers and administrators shall have access to support documentation and training. Administrators shall have access to configuration files and Interfaces, diagnostic instrumentation, logging, and alerting.
Non_Fun_Req_12	Compact hardware	Hardware installed in the home shall be as compact and robust as possible.
Non_Fun_Req_13	Energy-efficiency	Hardware installed in the home shall be as energy-efficient as possible.
Non_Fun_Req_14	Easy installation	Home hardware shall be easy and quickly installed.

**Table 2 sensors-19-00907-t002:** Selected sensors.

Sensor(s)	Manufacturer (Device Model)	EU Radio Frequency	Range	Power Supply	Battery Lifetime
Reed (Door/Window)	Fibaro (FGK-10x)	868.4 or 869.8 MHz	Up to 40m indoors, 50m outdoors	Battery: 3.6V DC ER14250 1/2 AA	12-24 months
Power (Wall plug)	Fibaro (FGWP-102)	868.4 or 869.8 MHz	Up to 40 m indoors, 50 m outdoors	230V AC, 50/60 Hz	N/A
Motion, Luminance, Temperature, Humidity	Aeotec (Multisensor 6 ZW100)	N/A	Up to 150 m outdoors	USB DC 5V or battery: 2xCR123 (3 V 1500 mAh)	Up to 24 months

**Table 3 sensors-19-00907-t003:** Validation results.

Dimension(Feature)	# of CEs	# of FPs	Error Rate	Accuracy	Margin of Error	Confidence Interval
duration per day	88	5	0,05682	0.94318	0,04837	[0.00845, 0.10519]
duration	952	20	0,02101	0,97899	0,00911	[0,01190, 0,03012]
time_off	107	2	0.01869	0.98131	0.02566	[0.00000, 0.04435]
time_on	105	1	0.00952	0.99048	0.01858	[0.00000, 0.02810]
time_reed_opened	846	2	0.00236	0.99764	0.00327	[0.00000, 0.00564]
time_reed_closed	849	0	0.00000	1.00000	0.00000	[0.00000, 0.00000]
movement_detected *	129	3	0,02326	0,97674	0,02601	[0.00000, 0.04926]
value	75	6	0.08000	0.92000	0.06140	[0.01860, 0.14140]

* A total number of movement events is 10,119. However, since movement events are periodic, we compare the total number of movement periods (three per day).

**Table 4 sensors-19-00907-t004:** End-user questionnaire and feedback.

Statement ID	Statement/Question	Average Agreement Level ^1^
S_01	Using the SmartHabits system was for me an enjoyable experience.	4.7
S_02	I consider the system is easy to use.	5
S_03	I would not need the support of a technical person to be able to use the system.	4
S_04	I think most people would quickly learn how to use the system.	5
S_05	I did not identify inconsistencies in this system.	5
S_06	I did not need to learn a lot before using the system.	5
S_07	I felt comfortable using the system.	4
S_08	I consider the user interface easy to use.	5
S_09	The user interface did not have too much content.	5
S_10	The user interface is well structured.	5
S_11	I would like to continue using this system.	4.3
S_12	I would recommend this system.	5
S_13	Using the system, I felt safer.	3.5
S_14	During the usage, I did not feel concerned about my privacy.	5

^1^ Level 1 being the lowest and level 5 being the highest agreement level related to the corresponding statement/question.

**Table 5 sensors-19-00907-t005:** Mapping of system requirements to system components and related statement(s).

Requirement ID	Requirement Short Name	System Components	Statement ID
Fun_Req_01	Learning support	ER, RRE	N/A
Fun_Req_02	Rules suggestion	RRE	S_04
Fun_Req_03	Rules management	RBAD, CP	N/A
Fun_Req_04	User management	PS	N/A
Fun_Req_05	Warning notifications	NM, CP, MC	S_04
Fun_Req_06	Home environment status	CP	S_01, S_04
Fun_Req_07	Home environment management	SP	N/A
Fun_Req_08	Local buffering	HGW	S_05
Fun_Req_09	Automatic start	HGW	S_05
Fun_Req_10	Notifications history	NM, CP, MC	S_04
Fun_Req_11	System notifications	NM, CP, MC, SMS, e-mail	S_05
Fun_Req_12	Adaptability	ER, RRE, RBAD	S_01
Non_Fun_Req_01	Configuration support	RBAD, CP, SP	S_01
Non_Fun_Req_02	Non-invasiveness	DGW, HGW	S_01, S_07, S_14
Non_Fun_Req_03	Zero-touch	DGW, HGW	S_01, S_02, S_03, S_06
Non_Fun_Req_04	Understandability	all	S_01, S_04, S_07
Non_Fun_Req_05	Affordability	Mostly Home-Sensing Platform hardware	S_12
Non_Fun_Req_06	Omnichannel notification delivery	NM	S_01, S_09
Non_Fun_Req_07	Appealing look-and-feel	CP, MC	S_01, S_02, S_08, S_09, S_10
Non_Fun_Req_08	Internationalization	CP, MC	S_04
Non_Fun_Req_09	Security	All, especially LS	S_14
Non_Fun_Req_10	Scalability	Cloud Platform with CP, SP	N/A
Non_Fun_Req_11	Supportability	CP, SP	S_04, S_06
Non_Fun_Req_12	Compact hardware	Home-Sensing Platform hardware	S_01
Non_Fun_Req_13	Energy-efficiency	Home-Sensing Platform hardware	S_11
Non_Fun_Req_14	Easy installation	DGW, HGW	S_02

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
