# Peer review of "The SmartHabits: An Intelligent Privacy-Aware Home Care Assistance System"

_sensors, 2019, doi:10.3390/s19040907_

Round 1

Reviewer 1 Report

This paper discusses detailed design and implementation of AAL (ambient assisted living) system using IoT devices. The implementation is pilot ready and was actually used in the evaluation in the field, which is a plus. The system design seems overall technically sound, and I think this paper has some value as an implementation study paper.

On the other hand, scientific contribution or novelty seems very low or is not enough explained. One major missing piece I saw is lack of discussion or evaluation machine learning part. I can imagine that a variety of data were collected by the system, but discussion about the details of machine learning method used, how the data is preprocessed, what features are originally considered, what feature selection method is applied, and justification for each is missing, which would not be convincing enough. In addition, the validity of derived rules etc. should have been evaluated systematically and/or quantitatively (i.e., not only by means of questionnaire).

Related to the above comment, it would be helpful for readers what kind of rules are considered and derived based on data. In particular, since authors collected real data through pilot testing, actual examples would be definitely interesting.

Regarding the system implementation, as authors mentioned, significant heterogeneity among IoT devices from different vendors. I wonder how the authors' system is extensible to support devices from different vendors. Such a discussion is important in practice. I am aware of one work that is designed for address the same issue. Perhaps authors could consider to cite it.

- Moazzami, Mohammad-Mahdi, et al. "SPOT: A smartphone-based platform to tackle heterogeneity in smart-home IoT systems." 2016 IEEE 3rd World Forum on Internet of Things (WF-IoT). IEEE, 2016.

Lastly, I saw English errors throughout the manuscript. Authors should perform careful proofreading before publication here or elsewhere.

Author Response

Response to Reviewer 1 Comments

Point 1: This paper discusses detailed design and implementation of AAL (ambient assisted living) system using IoT devices. The implementation is pilot ready and was actually used in the evaluation in the field, which is a plus. The system design seems overall technically sound, and I think this paper has some value as an implementation study paper. On the other hand, scientific contribution or novelty seems very low or is not enough explained. One major missing piece I saw is lack of discussion or evaluation machine learning part. I can imagine that a variety of data were collected by the system, but discussion about the details of machine learning method used, how the data is preprocessed, what features are originally considered, what feature selection method is applied, and justification for each is missing, which would not be convincing enough. In addition, the validity of derived rules etc. should have been evaluated systematically and/or quantitatively (i.e., not only by means of questionnaire).

Response 1: Thank you for your valuable comments, we have added additional information in our manuscript. In the entirely new section 3.5. we discuss the SmartHabits anomaly detection process in much more details. The new section includes the information how the data was collected and pre-processed, what features are considered, what feature method was applied and how the machine learning was used for anomaly detection within SmartHabits. For the complete view, the existing figure showing basic rule-based anomaly detection data flow was moved from section 3.4.2. to the new section 3.5. Additional validation results were added in section 5. together with the figure showing the average number of learned patterns per sensor.

Point 2: Related to the above comment, it would be helpful for readers what kind of rules are considered and derived based on data. In particular, since authors collected real data through pilot testing, actual examples would be definitely interesting.

Response 2: Existing figure 6 (the rule details view) shows one example of the rule in section 3.4.3. Additional (new) information related to the rules was added in the new section 3.5. while an additional example of warning message (related to the rule) was added in section 4.1.2. alongside previously mentioned ones.

Point 3: Regarding the system implementation, as authors mentioned, significant heterogeneity among IoT devices from different vendors. I wonder how the authors' system is extensible to support devices from different vendors. Such a discussion is important in practice. I am aware of one work that is designed for address the same issue. Perhaps authors could consider to cite it. Moazzami, Mohammad-Mahdi, et al. "SPOT: A smartphone-based platform to tackle heterogeneity in smart-home IoT systems." 2016 IEEE 3rd World Forum on Internet of Things (WF-IoT). IEEE, 2016.

Response 3: The information on extensibility was added in two places. The information on the data model design that is based on the well-established ontologies and guidelines was added in section 3.4.2. At the end in the section 4.3. the information on extensibility of the SmartHabits system was given confirming openness to extensions and existing practices. Thank you for the paper which we consider relevant and cite it, together with one more additional paper.

Point 4: Lastly, I saw English errors throughout the manuscript. Authors should perform careful proofreading before publication here or elsewhere.

Response 4: We have performed extensive proofreading’s and as a result applied English improvements throughout the manuscript. We have focused on grammar improvements as well as readability, fluency and clarity improvements. Particular focus was put on the abstract which we believe is now more fluent.

Reviewer 2 Report

SmartHabits system is described in this paper. 

Such technology is aligned with the most recent development in AAL systems field.

From the technological point of view the project is well articulated and illustrated with enough details: Zwave environmental wireless sensors, Raspberry Pi Single Board Computer, Cloud Paradigm, Pattern Recogniction and Anomaly Detections.

However, some aspects of the work can be improved - minor revisions could concern the following points:

1. great attention is given to the issues of privacy and data security but it is not very clear how the communication between the Zwave sensors and the Raspberry Pi is protected - this seems to be the weak link in the IoT chain designed.

2. the algorithms at the base of the anomaly detection engine and mostly of the pattern recognition engine are not explained but only hinted at and this confers vagueness to the description.

3. the environmental sensors used are rather generic and it is difficult understand what useful information the system can generate - the utility of the system as a whole is not well focused.

4. the system provides few functionalities (users state that more functions related to dangerous conditions would be needed: flooding, gas leaks, falls, ...) - it is possible to create rules to define the anomalous situation of interest (RBAD) but this approach covers solved predictable situations, it also requires a configuration activity - the ability to automatically generate the rules (RRE) is more interesting but this possibility is not well illustrated.

5. no examples of positive results produced by the system or numerical data that demonstrate the validity of the approach adopted are discussed.

Author Response

Response to Reviewer 2 Comments

Point 1: Great attention is given to the issues of privacy and data security but it is not very clear how the communication between the Zwave sensors and the Raspberry Pi is protected - this seems to be the weak link in the IoT chain designed.

Response 1: Thank you for your insightful comments. Information explaining the sensors-RPi Z-Wave communication and protection thereof was added in section 4.3. (below the table). New content includes the information that sensors used in the pilot support Z-Wave network security mode with AES-128 encryption.

Point 2: the algorithms at the base of the anomaly detection engine and mostly of the pattern recognition engine are not explained but only hinted at and this confers vagueness to the description.

Response 2: We have added a completely new section 3.5. where we discuss the SmartHabits anomaly detection process in much more details. The new section includes the information on all steps in the used process with special focus on algorithms and pattern recognition engine.

Point 3: the environmental sensors used are rather generic and it is difficult understand what useful information the system can generate - the utility of the system as a whole is not well focused.

Response 3: Additional information was added in section 3.5. and the update of section 5. At the beginning of section 3.5. the context, goal, and pre-conditions of the SmartHabits anomaly detection process are elaborated.

Point 4: the system provides few functionalities (users state that more functions related to dangerous conditions would be needed: flooding, gas leaks, falls, ...) - it is possible to create rules to define the anomalous situation of interest (RBAD) but this approach covers solved predictable situations, it also requires a configuration activity - the ability to automatically generate the rules (RRE) is more interesting but this possibility is not well illustrated.

Response 4: More detailed information of the whole rule generating process was added within the new section 3.5. In the process, the existing figure showing basic rule-based anomaly detection data flow was moved from section 3.4.2. to the new section 3.5.

Point 5: no examples of positive results produced by the system or numerical data that demonstrate the validity of the approach adopted are discussed.

Response 5: Additional results were presented in section 5. including figure containing the average number of learned patterns per sensor.
